# Dynamic dichotomy of accumbal population activity underlies cocaine sensitization

**Ruud van Zessen[1], Yue Li[1], Lucile Marion-Poll[1], Nicolas Hulo[2], Jérôme Flakowski[1], Christian Lüscher[1,3]***

[1]Department of Basic Neurosciences, Faculty of Medicine, University of Geneva, Geneva, Switzerland; [2]Institute of Genetics and Genomics of Geneva (IGE3), University of Geneva, Geneva, Switzerland; [3]Clinic of Neurology, Dept. of Clinical Neurosciences, Geneva University Hospital, Geneva, Switzerland

**Abstract** Locomotor sensitization (LS) is an early behavioral adaptation to addictive drugs, driven by the increase of dopamine in the Nucleus Accumbens (NAc). However, the effect on accumbal population activity remains elusive. Here, we used single-cell calcium imaging in mice to record the activity of dopamine-1-receptor (D1R) and dopamine-2-receptor (D2R) expressing spiny projection neurons (SPNs) during cocaine LS. Acute exposure to cocaine elevated D1R SPN activity and reduced D2R SPN activity, albeit with high variability between neurons. During LS, the number of D1R and D2R neurons responding in opposite directions increased. Moreover, preventing LS by inhibition of the ERK signaling pathway decreased the number of cocaine responsive D1R SPNs, but had little effect on D2R SPNs. These results indicate that accumbal population dichotomy is dynamic and contains a subgroup of D1R SPNs that eventually drives LS. Insights into the drug-related activity dynamics provides a foundation for understanding the circuit-level addiction pathogenesis.

*For correspondence:
Christian.Luscher@unige.ch

**Competing interest:** The authors declare that no competing interests exist.

## Introduction

Drug addiction is a chronic relapsing disorder without cure, in part because of incomplete pathophysiological understanding of the disorder. While addictive drugs have distinct molecular targets, their actions converge through the increase of dopamine (DA) levels in the Nucleus Accumbens (NAc) (*Di Chiara and Imperato, 1988*). Starting with the first drug exposure many adaptations occur, providing a foundation from which addiction can develop. However, it remains elusive how these initial drug exposures affect the output activity of the NAc.

In rodent models of early drug adaptations, exposure to psychostimulants drives locomotion. This effect becomes more pronounced upon repetitive exposure to the same dose, and is termed locomotor sensitization (LS). It is part of a larger collection of drug-induced behaviors that sensitize over drug exposures (*Robinson and Berridge, 1993*). LS is dependent on the timing between drug exposures, as well as on the environment in which the animal receives the drugs (*Robinson et al., 1998; Valjent et al., 2010*). It has been argued that LS may reflect the incentive saliency of the addictive drug; however, its existence in humans is still debated (*Robinson and Berridge, 2008*).

Psychostimulant-induced hyperlocomotion is dependent on NAc dopamine and glutamate signaling (*Kelly and Iversen, 1976; Pulvirenti et al., 1991*). The NAc consists mainly of spiny projection neurons (SPNs) that can be characterized by their expression of different subtypes of DA receptors. One class expresses Dopamine-1 receptors (D1Rs) that activate $G_{s/olf}$ coupled G-proteins, while the other expresses dopamine-2 receptors (D2Rs) that are $G_{i/o}$ coupled (*Gerfen and Surmeier, 2011*). Among the intracellular signaling cascades, the activation of the extracellular-signaling related kinase

(ERK) pathway in D1R SPNs is of particular interest. When this pathway is blocked, LS fails to develop (*Valjent et al., 2006*). When signaling through this ERK pathway is paired with afferent excitatory transmission, synaptic potentiation of glutamatergic synapses is induced, resembling NMDAR-dependent LTP. Following withdrawal of the first cocaine exposure, afferents from the medial prefrontal cortex (mPFC) and ventral hippocampus onto NAc D1R SPNs, but not on D2R SPNs, undergo this synaptic plasticity (*Pascoli et al., 2011*; *Pascoli et al., 2014*).

Several recent studies have established a causal role for this enhanced synaptic transmission onto D1R SPNs in early drug-adaptive behavior. For example, reversing cocaine-evoked synaptic potentiation abolishes LS without affecting the initial locomotor response (*Pascoli et al., 2011*; *Creed et al., 2015*). Moreover, optogenetic inhibition of the D1R-SPNs that have been exposed to a DA transient evoked by a single injection of cocaine abolishes LS (*Lee et al., 2017*).

While much is known about the molecular and cellular determinants of LS, it remains unknown how repeated exposure to cocaine affects the neuronal population response in vivo. During operant behavior, cocaine and cocaine-paired actions elicit very heterogenous responses in NAc neurons (*Carelli et al., 1993*; *Peoples et al., 1998*), and bulk calcium imaging indicate differential cocaine responses between NAc SPN subclasses (*Calipari et al., 2016*). To characterize in vivo cocaine responses on a single neuronal level during behavioral sensitization we imaged the calcium dynamics in identified populations of SPNs with a miniature microscope. We found that after initial cocaine exposure, a fraction of D1R and D2R-SPN show divergent and opposing activity. Importantly, this dichotomous response was enhanced during the expression of LS, with more D1R SPNs elevating their activity and a larger proportion of D2R SPNs reducing their activity. While a fraction of accumbal SPNs were movement-responsive, these were largely different neurons from those that were cocaine responsive, and their coding did not change during LS. Finally, pharmacologically inhibiting ERK signaling was sufficient to reverse cocaine induced D1R SPN activity changes and block the development of LS.

## Results

### Diverse activity responses to initial cocaine exposure

We imaged calcium signals in identified accumbal neurons of freely moving mice as a proxy for their activity. To target D1R SPNs we used D1R-cre transgenic mice, and for D2R SPNs we used D2R-cre or A2a-cre transgenic mice. We injected animals with a floxed viral construct into the NAc to express GCaMP6f. We then placed a GRIN lens above the infected area, connected a miniaturized microscope (*Figure 1A*, *Figure 1—figure supplement 1A,E*) and on three different sessions imaged four periods of 5 min around an intraperitoneal (I.P.) injection. This allowed us to record calcium transients in response to one saline and two cocaine (20 mg/kg, I.P.) injections where the latter were separated by 1 week (*Figure 1B*). We calculated the locomotor sensitization index as the ratio of the distance moved between these two cocaine injections. Cocaine elicited a locomotor response that was enhanced after the second cocaine injection (*Figure 1C*, *Figure 1—figure supplement 2A*). We then extracted individual neuronal calcium signals (*Figure 1D*) and identified calcium transients by thresholding the derivative of the temporal trace, isolating, and then fitting, the individual transients (see Materials and methods). The frequency of calcium transients was highly variable, even between neurons of the same animal (*Figure 1E–H*, *Figure 1—figure supplement 2B*). We focused on the change in frequency from the last 5 min before, to the first 5 min after, cocaine administration. These periods showed high divergence in behavioral (*Figure 1—figure supplement 2A*) as well as neuronal (*Figure 1—figure supplement 2B*) responses. Overall, there was an elevation in D1R SPN activity and a reduction in D2R SPN activity (*Figure 1I*), an effect that was not present in control animals that only received repeated saline injections (*Figure 1J*). These activity changes likely reflect alterations in excitability due to G proteins signaling downstream of DA receptors, in combination with excitatory inputs (*Nicola, 2007*). Thus, while individual neurons show diverse responses, on average SPNs fell into two dichotomous classes: D1R SPNs show an elevated activity, while D2R SPNs show a reduced activity following cocaine exposure.

### Recruitment of cocaine-responsive NAc neurons with LS

Next, we monitored the impact of the second, sensitizing, cocaine injection on calcium transients in the two SPN populations. We observed a high variability of response to cocaine across D1R or D2R

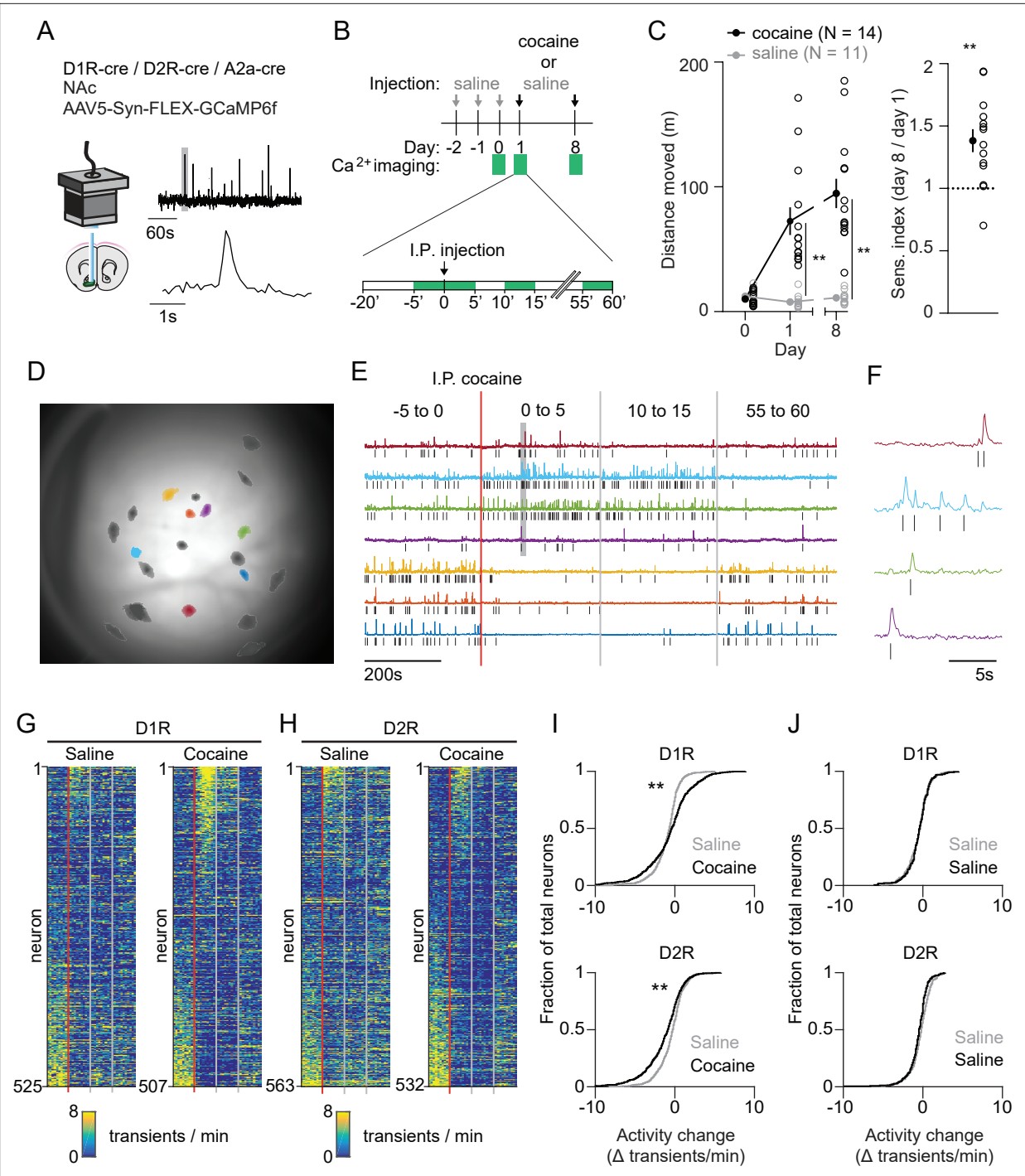

**Figure 1.** Acute changes in accumbal neuronal activity in response to cocaine. (**A**) Schematic of calcium imaging setup. Inset graph shows calcium recording trace, gray line indicates zoomed area in lower graph. Top trace horizontal bar is 60 s, bottom trace horizontal bar is 1 s. (**B**) Schematic of behavioral setup. Green squares indicate recording days (top) and 5-min recording windows (bottom). (**C**) Left panel: distance moved 1 hr after I.P. injection in mice that either receive cocaine (N = 14) or saline control (N = 11) injections, for two sessions separated by 1 week. Average responses (mean ± S.E.M.): cocaine group: day 1: 72.5 ± 11.2 m, day 8: 94.8 ± 11.8 m. Saline group: day 1: 7.8 ± 1.6 m, day 8: 11.0 ± 1.5 m. Repeated measures ANOVA: Days x Treatment: $F_{(2,46)}$ = 30.74, Bonferroni-corrected post hoc T-test: ** = p < 0.01. Right panel: sensitization index in cocaine treated animals. Average response (mean ± S.E.M.): sensitization index 1.38 ± 0.09. Paired T-test: p = 1.2 x 10$^{-3}$. ** = p < 0.01 (**D**) Field of view of a single animal (D1R-cre), overlaid with a subset of neurons recorded during a single cocaine-treated session. (**E**) Temporal trace (CNMF-E label 'C_raw') of highlighted neurons in panel (**D**), showing diverse activity responses to first cocaine injection within the same D1R-cre mouse. Raw traces are rescaled to their own maximum. Tick marks indicate detected calcium transients. (**F**) Zoom of highlighted area (gray box) in panel (**F**), showing calcium signals over time.

*Figure 1 continued on next page*

*Figure 1 continued*

Tick marks indicate detected calcium transients. (**G**) Calcium transient frequency heatmap of all neurons recorded in D1R-cre cocaine treated animals during either saline (left) or cocaine (right) treatment. Data is sorted by amplitude of change in activity. N = 7 animals. (**H**) Same as panel (**G**) but for D2R-cre cocaine treated animals. N = 7 animals. (**I**) Top panel: Cumulative distribution of activity change (−5 to 0 vs 0 to 5) minutes after I.P. injection in D1R neurons of the cocaine-treated groups around either saline (grey) or cocaine (black) treatment. Saline: n = 525 neurons, cocaine n = 507 neurons. Two-sample Kolmogorov–Smirnov test: $p = 1.3 \times 10^{-10}$. Bottom panel: same but for D2R neurons. Saline: n = 563 neurons, cocaine n = 532 neurons. Kolmogorov-Smirnov test: $p = 1.3 \times 10^{-07}$. ** = $p < 0.01$ (**J**) Top panel: cumulative distribution of activity change (−5 to 0 vs. 0 to 5 min around I.P. injection) in D1R neurons of control animals treated only with saline on 2 separate days. First saline: n = 273 neurons, second saline: n = 235 neurons. Kolmogorov-Smirnov test: $p = 1$. Bottom panel: same but for D2R neurons. First saline: n = 231 neurons, second saline: n = 256 neurons. Kolmogorov-Smirnov test: $p = 0.054$. Kolmogorov-Smirnov tests are corrected for multiple comparisons.

The online version of this article includes the following figure supplement(s) for figure 1:

**Figure supplement 1.** Schematic overview of fiber / GRIN placement for different experimental groups.

**Figure supplement 2.** Diverse movement and calcium activity responses after cocaine.

neurons for all animals (*Figure 2—figure supplement 1A*). To identify neurons with the most robust activity change we applied a bootstrapping method on all recording sessions (*Rozeske et al., 2018*). Per neuron, we again isolated the calcium transient frequency change between the last 5 min before injection and the first 5 min after injection. We then compared this to 10,000 activity sets from the same neuron, where the sequence of intervals between transients were shuffled. Activity changes in neurons that fall within the top percentile of the shuffled data were identified as cocaine-responsive neurons (*Figure 2A*). In both D1R and D2R SPN populations, this method identified responsive neurons that either elevate or reduce their activity to cocaine (*Figure 2B–C*). We then asked whether the number of these responsive neurons would change after repeated cocaine exposure. We quantified the fraction of neurons that were responsive on each recording day in cocaine treated and control groups. D1R and D2R responses to cocaine were different (*Figure 2D*). In D1R neurons, the fraction of cocaine responsive neurons increased between saline and cocaine days, especially the fraction of neurons elevating their activity. Importantly, this fraction increased over the sensitization (*Figure 2D*). In D2R neurons the opposite occurred, as the number of neurons with reduced activity was increased between saline and the first cocaine exposure, and further increased during the second cocaine treatment (*Figure 2D*). To further validate these findings, we performed a complementary analysis, considering that some neurons are recorded from the same animal. The linear mixed model (see Materials and methods) also shows that responses are significantly different over cocaine days for D1R and D2R neurons (*Figure 2—figure supplement 1B*). Taken together, we identified cocaine-responsive neurons and observed that the divergent response of D1R and D2R SPNs to the first cocaine injection was more pronounced after a sensitizing second cocaine injection.

## Characterization of cocaine-responsive SPN subgroups

We next sought to characterize the activity of these cocaine-responsive neurons over cocaine exposures. To follow neurons over sessions we used a probabilistic cell registration method (*Sheintuch et al., 2017*; *Figure 3A and B*). We were able to track around 56 % of NAc SPNs (581/1039 neurons, 251/507 D1R, 330/532 D2R) over the two sessions. If we consider all tracked neurons, we find that for a given SPN, cocaine responses were consistent over days, particularly in D1R SPNs (*Figure 3C and F*). We found that in both D1R and D2R SPN populations, the cocaine response to the second cocaine injection contained both neurons that had previously been responsive, as well as newly recruited neurons (*Figure 3D and G*). We then asked whether the activity within these subgroups also changed with sensitization. For neurons that were not considered statistically responsive during the first cocaine injection, but then showed significant responsiveness only during the second injection, activity was significantly elevated in D1R SPNs (*Figure 3E*, left panel), and reduced in D2R SPNs (*Figure 3H*, left panel). By contrast, neurons that responded to both injections showed similar activity in the two conditions (*Figure 3E* right panel, *Figure 3H*, right panel). These results indicate that cocaine responses in D1R and D2R SPNs are conserved within the same neuron between cocaine exposures. Sensitization was associated with an increasing number of responding neurons, further enhancing the D1R-D2R dichotomy.

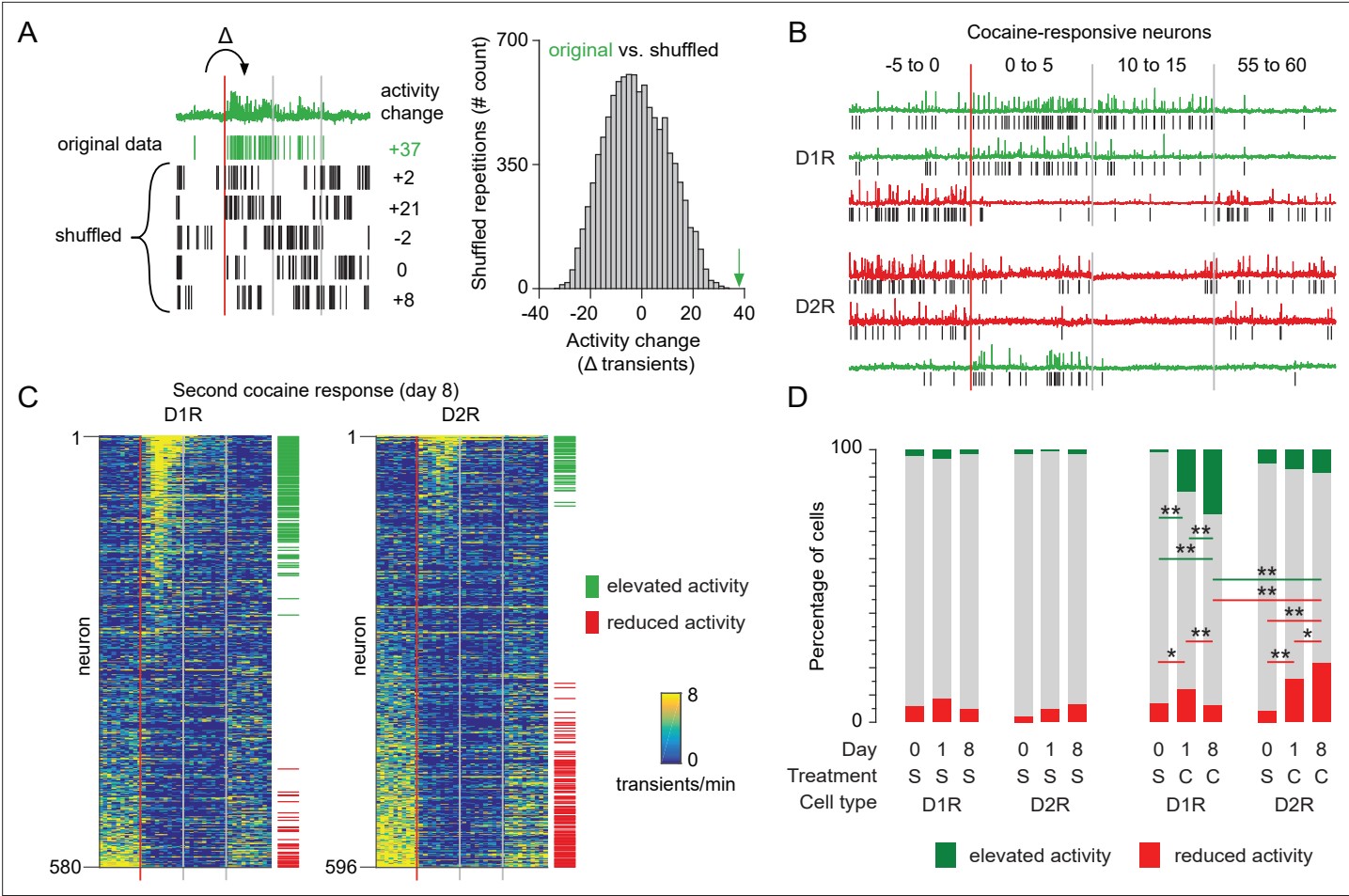

**Figure 2.** Recruitment of cocaine-responsive neurons during cocaine sensitization. (**A**) Schematic overview of methodology to identify responsive neurons. Calcium transients are detected from traces (green trace shows raw trace, green ticks show calcium transients), and activity change around cocaine injection is compared to shuffled sequences of the same inter-transient intervals (bottom left and histogram). Vertical lines indicate edges of five-minute recording windows, red line indicates cocaine injection. Green arrow indicates non-shuffled result. (**B**) Calcium traces of example D1R (top three) or D2R (bottom three) neurons that either show significant increased (green) or decreased (red) activity to cocaine. Vertical red line indicates cocaine injection moment, red and gray lines indicate edges of recording periods. Black ticks indicate detected calcium transients. (**C**) Calcium transient frequency heatmap of all neurons recorded around second cocaine injection in either D1R-cre mice (left, n = 580 neurons from seven animals) or D2R-cre mice (right, n = 596 neurons from seven animals). Neurons are sorted by their response to cocaine, green and red ticks indicate significantly responsive neurons. (**D**) Fraction of responsive neurons among D1R-cre and D2R-cre neurons in saline control or cocaine-treated animals. S = saline. C = cocaine. D1R control n = 273/235/229 neurons (day 0 /day 1 /day 8) from 6 mice. D2R control n = 231/256/231 neurons from 5 mice. D1R cocaine n = 525/507/580 neurons from 7 mice. D2R cocaine n = 563/532/596 neurons from 7 mice. Fishers Exact test: D1R cocaine group, elevated activity: day 0 vs day 1: p = 5.9 x 10$^{-19}$, day 1 vs day 8: p = 1.9 x 10$^{-3}$, day 0 vs day 8: 1.5 x 10$^{-34}$. Reduced activity: day 0 vs day 1: p = 1.6 x 10$^{-2}$, day 1 vs day 8: p = 1.7 x 10$^{-3}$. D2R cocaine group reduced activity: day 0 vs day 1: p = 3.1 x 10$^{-10}$. day 1 vs day 8: p = 4.4 x 10$^{-2}$. Day 0 vs day 8: p = 3.4 x 10$^{-19}$. All other intra-group comparisons p > 0.05. Day 8 cocaine, D1R vs D2R: elevated activity, p = 9.0 x 10$^{-13}$, reduced activity, p = 4.1 x 10$^{-15}$. * = p < 0.05, ** = p < 0.01.

The online version of this article includes the following figure supplement(s) for figure 2:

**Figure supplement 1.** Analysis of the cocaine response, accounting for individual animals.

## Accumbal movement-coding cannot explain LS

While activity of accumbal SPNs have primarily been involved in reward-related behaviors, along with neurons in the dorsal striatum, they also code for actions (*Nicola, 2007*; *Klaus et al., 2017*). Since the sensitization is behaviorally expressed as increased movement, we explored the possibility that altered population activities might simply reflect the encoding of movement. To this end, we aligned the onset of calcium transients with the concomitant velocity, implementing an approach previously used in the dorsal striatum (*Parker et al., 2018*; *Figure 4A*). We then identified neurons with the strongest correlation to changes in movement (*Figure 4B*), and termed these velocity-responsive

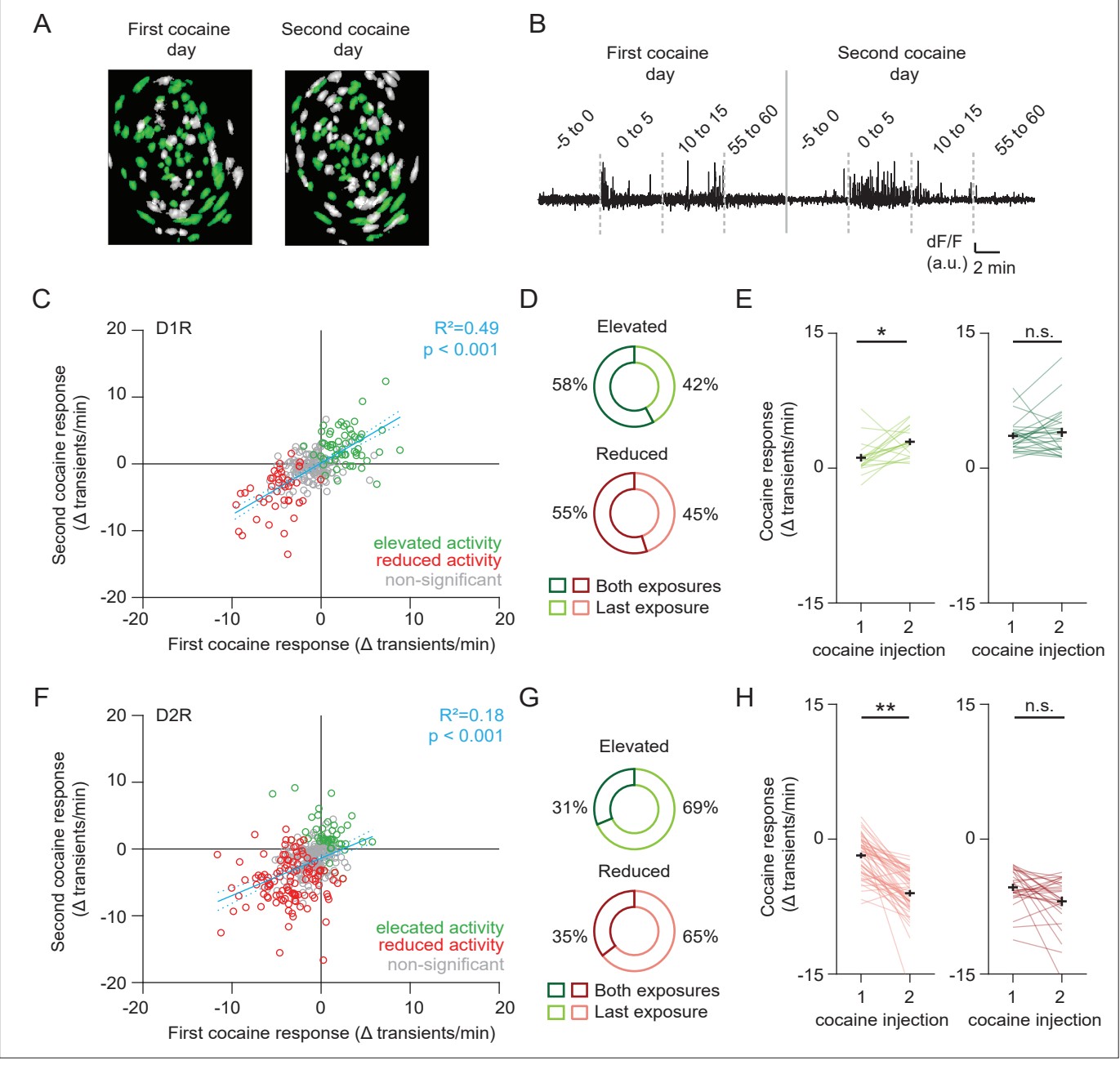

**Figure 3.** Profiling cocaine responsive neurons over cocaine treatments. (**A**) Example of identified neurons on two different sessions in the same animal. Green indicates neurons present on both session (see Methods). (**B**) Example trace of a single neuron that was present during both cocaine-treatment days. (**C**) Scatter plot showing activity change of D1R neurons during the first and second cocaine injections. Neurons that are cocaine responsive on at least one session are highlighted (green – increased activity, red decreased activity). Blue line shows linear correlation across all neurons (dotted line is 95% C.I.) Pearson's $r = 0.77$, $p = 2.8 \times 10^{-38}$, $n = 251$ neurons from 7 mice. (**D**) Pie chart showing fraction of D1R neurons with increased activity on second cocaine exposure day, that were either responsive during both sessions (dark colors), or non-significantly responding during the first cocaine exposure (light colors). Neurons per group: elevated activity: both exposures $n = 26/45$, last exposure $n = 19/45$. Reduced activity: both exposures $n = 11/20$, last exposure $n = 9/20$. (**E**) Activity change between cocaine-treated days among D1R neurons that increase their activity following cocaine: those that are non-responsive during the first cocaine exposure and are responsive during the second exposure (left), and neurons that are responsive to cocaine on both days (right). Wilcoxon signed-rank test: left panel: $n = 19$, $p = 0.014$, right panel: $n = 26$, $p = 0.80$. * = $p < 0.05$. (**F**) Same as (**C**) but for D2R neurons that decrease their activity following cocaine. Pearson's $r = 0.56$, $p = 1.4 \times 10^{-15}$, $n = 330$ neurons from 7 mice. (**G**) Same as (**D**) but for D2R neurons that reduce their activity following cocaine. Neurons per group: elevated activity: both exposures $n = 10/32$, last exposure $n = 22/32$. Reduced activity: both exposures $n = 28/79$, last exposure $n = 51/79$. (**H**) Same as (**E**) but for D2R neurons. Wilcoxon signed-rank test: left panel: $n = 51$, $p = 2.3 \times 10^{-9}$, right panel: $n = 28$, $p = 0.08$. ** = $p < 0.01$.

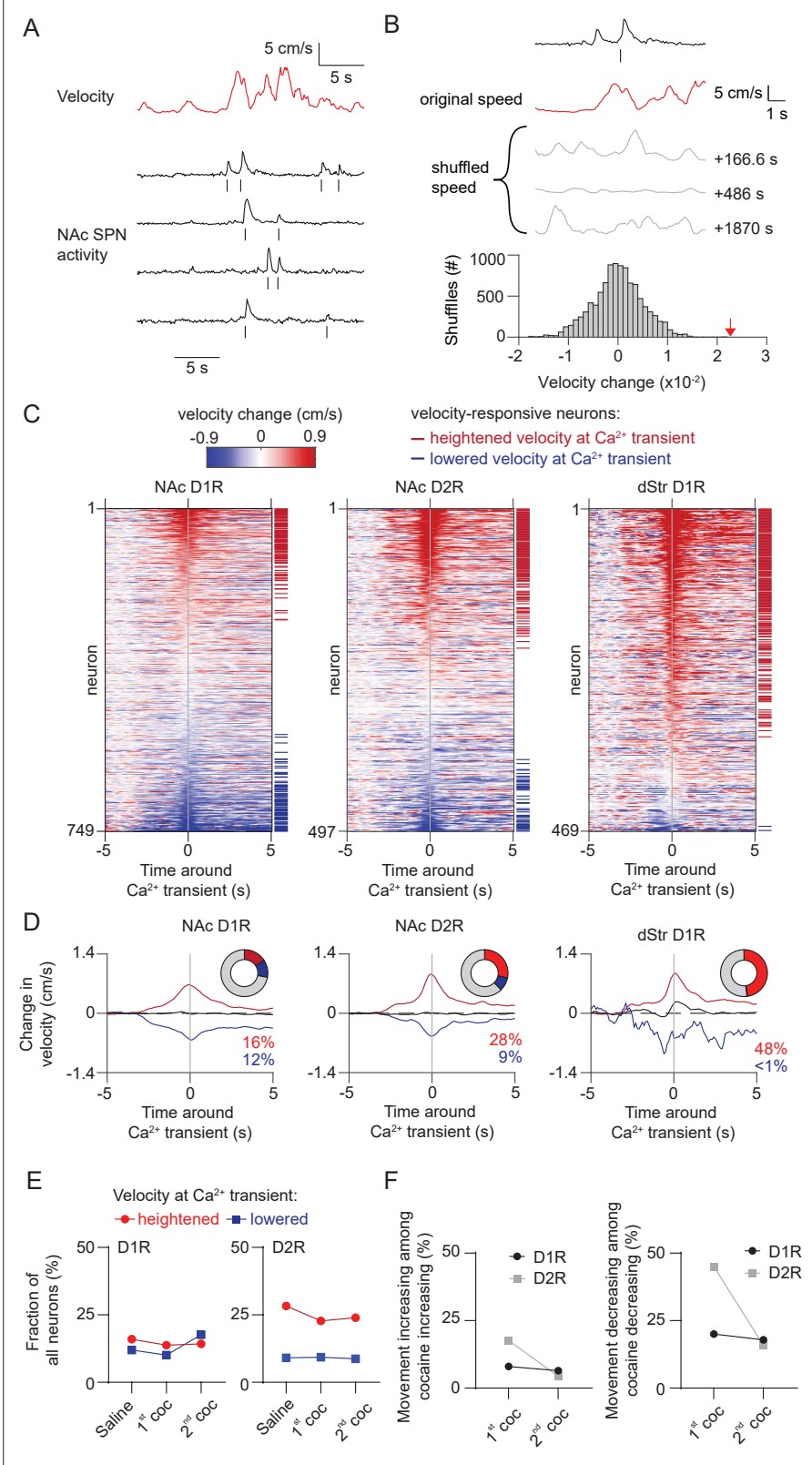

**Figure 4.** Movement-related activity of NAc neurons does not change during cocaine sensitization. (**A**) Example trace showing velocity over time and the activity of D1R neurons recorded during a saline-treated session. Ticks below each trace indicate detected calcium transients. (**B**) Schematic overview of methodology to identify neurons correlated to velocity. Calcium transients are detected from traces and velocity change around each transient

*Figure 4 continued on next page*

*Figure 4 continued*

is computed, and then averaged per neuron. Alignment is then compared to temporally shifted shuffled data (where the velocity is offset by a random time period) to identify movement-correlating neurons (see Materials and methods). Gray lines indicate examples of temporally shifted data, and histogram shows distribution of this data. Red arrow indicates non-shuffled result. (**C**) Heatplot showing average velocity change aligned to calcium transients in either NAc D1R neurons (left), NAc D2R neurons (middle) or dStr D1R neurons (right) under saline injection conditions. Neurons are sorted by their movement response amplitude. Horizontal red and blue lines indicate significantly velocity-responsive neurons. (**D**) Average activity over time of neurons correlated with either heightened (red), non-significant (black) or lowered (blue) movement at the time of the calcium transient under saline conditions. Pie chart insets shows fraction of total neurons with significant response. (**E**) Fraction of neurons correlated to movement over days. (**F**) Left: fraction of neurons correlated to increases of movement among cocaine-responsive neurons with elevated activity. Right: fraction of neurons correlated to decreases of movement among cocaine-responsive neurons with reduced activity.

The online version of this article includes the following figure supplement(s) for figure 4:

**Figure supplement 1.** Movement around calcium transients in all recorded neurons over days.

neurons. To characterize velocity responsiveness under naive conditions we initially performed these analyses on the day of saline treatment. For comparison, we also performed similar recordings in a new cohort of D1R-cre animals while imaging in the dorsal striatum (*Figure 1—figure supplement 1B*). We indeed found neurons among NAc D1R and D2R SPNs, as well as dStr D1R SPNs, whose activity was correlated with movement (*Figure 4C*). For velocity-responsive neurons of the NAc and dStr similar temporal velocity dynamics were observed (*Figure 4D*). However, of all NAc D1R SPNs only a modest (~28%) fraction was associated with movement. SPN activity also correlated with heightened and lowered velocity regardless of the cell type, while in the dorsal striatum calcium transients were almost exclusively aligned to heightened velocity (*Figure 4D*). We then asked whether the amount of velocity-responding neurons would increase over LS. This was not the case, as the fraction of neurons that were velocity-responsive was unchanged over days (*Figure 4E*). Moreover, when not taking our classification of velocity-responsive neurons into account and instead looking at all NAc neurons, there were also no clear changes in velocity alignment over LS (*Figure 4—figure supplement 1*). However, it is still possible that the number of velocity-responsive neurons among cocaine-responsive neurons changes during sensitization, and would potentially explain their cocaine responsiveness. Particularly, velocity-responsive neurons with heightened velocity at calcium transients among D1R SPNs that elevate their activity to cocaine exposure could potentially explain their changes in cocaine-responding over LS. And the opposite would explain D2R SPNs evolution over LS, namely an increase of velocity-responsive neurons with lowered velocity among cocaine-responsive neurons that reduce activity. However, contrary to this hypothesis, we found no increase in velocity-responsive neurons in either group of cocaine-responsive neurons, neither among D1R or D2R neurons (*Figure 4F*). We thus confirm that under basal conditions in the NAc fewer neurons are movement-correlated than in the dStr. Interestingly, in the NAc roughly an equal number of D1R-SPNs were aligned to heightened or lowered velocity, while in the dorsal striatum only alignment to heightened velocity was observed. However, the number of velocity-responsive neurons did not change with LS, in line with the idea in dorsal striatum that movement-related activity of SPNs is obfuscated under cocaine (*Barbera et al., 2016*). Finally, the fraction of velocity-responsive neurons among cocaine-responsive neurons was also not increased, suggesting that the majority of cocaine-responsive D1R and D2R-SPNs did not code for movement per se.

## Cocaine-induced NAc dopamine transients remain stable with LS

We next set out to investigate the causes underlying the recruitment of cocaine responsive neurons. Since acute cocaine-evoked locomotion depends on dopamine (*Kelly and Iversen, 1976*), sensitization may simply reflect enhanced accumbal dopamine transients. To test this possibility, we measured dopamine levels using the fluorescent dopamine sensor dLight 1.1 (*Patriarchi et al., 2018*). Wild-type animals were injected with an AAV containing a CAG-driven dLight 1.1 into the NAc, and an optic fiber was placed above the area (*Figure 5A*, *Figure 1—figure supplement 1C*). Like in earlier experiments, animals were exposed to saline and cocaine on subsequent days, followed by a second cocaine injection 1 week later. Robust DA elevations were observed after the first and second injections of cocaine

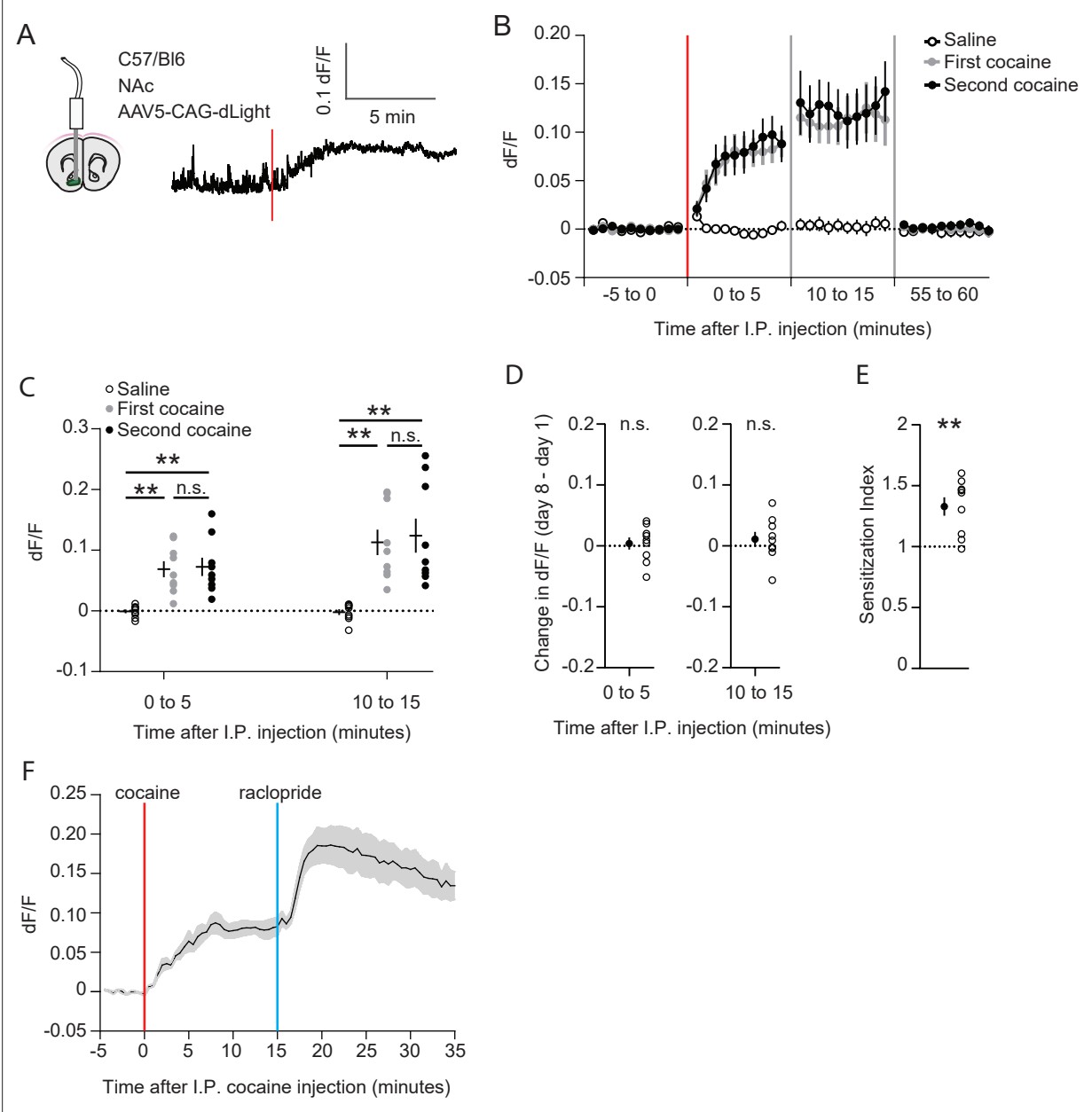

**Figure 5.** Cocaine-evoked NAc dopamine levels remain stable during cocaine sensitization. (**A**) Experimental setup. Wild-type mice were injected with an AAV9 CAG-dLight construct in the NAc, and an optic fiber was placed above the area. Trace represents dLight fluorescence over time. Vertical red line indicates moment of cocaine injection. (**B**) Average dF/F over time around I.P. injection of either saline or cocaine. Red line indicates cocaine injection. N = 9 mice. (**C**) Average dF/F over 5-min periods. Greenhouse-Geisser corrected Repeated measures ANOVA F (1.100, 8.802) = 8.816, p = 0.015. Bonferroni-corrected two-tailed paired students T-test: 0–5 min: saline vs first cocaine: p = 1.6 x 10⁻³, saline vs. second cocaine: p = 3.5 x 10⁻³, first cocaine vs second cocaine p > 0.99. 10–15 minutes: saline vs first cocaine: p = 1.7 x 10⁻³, saline vs second cocaine: p = 6.2 x 10⁻³, first cocaine vs second cocaine: p > 0.99. ** = p < 0.01. (**D**) Within animal change of five-minute average dF/F between cocaine exposures. Paired students T-test: 0–5 minutes: p = 0.71, 10–15 min: p = 0.39. (**E**) Sensitization index of the same animals shown in panels B-D. Paired students T-test: p = 2.5 x 10⁻³. ** p < 0.01. (**F**) Average dF/F over time around I.P. injection of sequential cocaine (20 mg/kg, I.P.) and D2 antagonist raclopride (0.1 mg/kg, I.P.). N = 5 mice.

(*Figure 5B*). Fluorescence levels increased in the first 5 min and remained at a high plateau between 10 and 15 min (*Figure 5C*), following a time course like the drug-evoked locomotion (*Figure 1— figure supplement 2A*). However, the fluorescence levels were indistinguishable for the two injections (*Figure 5C*), even when comparing the fluorescence in the same animal (*Figure 5D*). Importantly, these same animals show clear locomotor sensitization (*Figure 5E*), like the previous experiment (*Figure 1C*). To rule out dLight saturation, we injected the D2R antagonist raclopride (0.1 mg/kg, I.P.)

at the peak of the cocaine response. Since midbrain DA neuron firing is lowered due to D2R auto-receptors (*Wei et al., 2018*), blocking these receptors will enhance activity and DA release, which is faithfully reflected by dLight1.1 (*Figure 5F*). Thus, while cocaine acutely increases NAc DA levels, locomotor sensitization and the increase in population response cannot simply be explained by higher dopamine levels.

## Inhibition of ERK signaling prevents LS and recruitment of cocaine-responsive SPNs

Previous studies show that blocking the ERK signaling pathway abolishes cocaine-induced synaptic plasticity at excitatory inputs to NAc D1R SPNs and prevents locomotor sensitization (*Valjent et al.,*

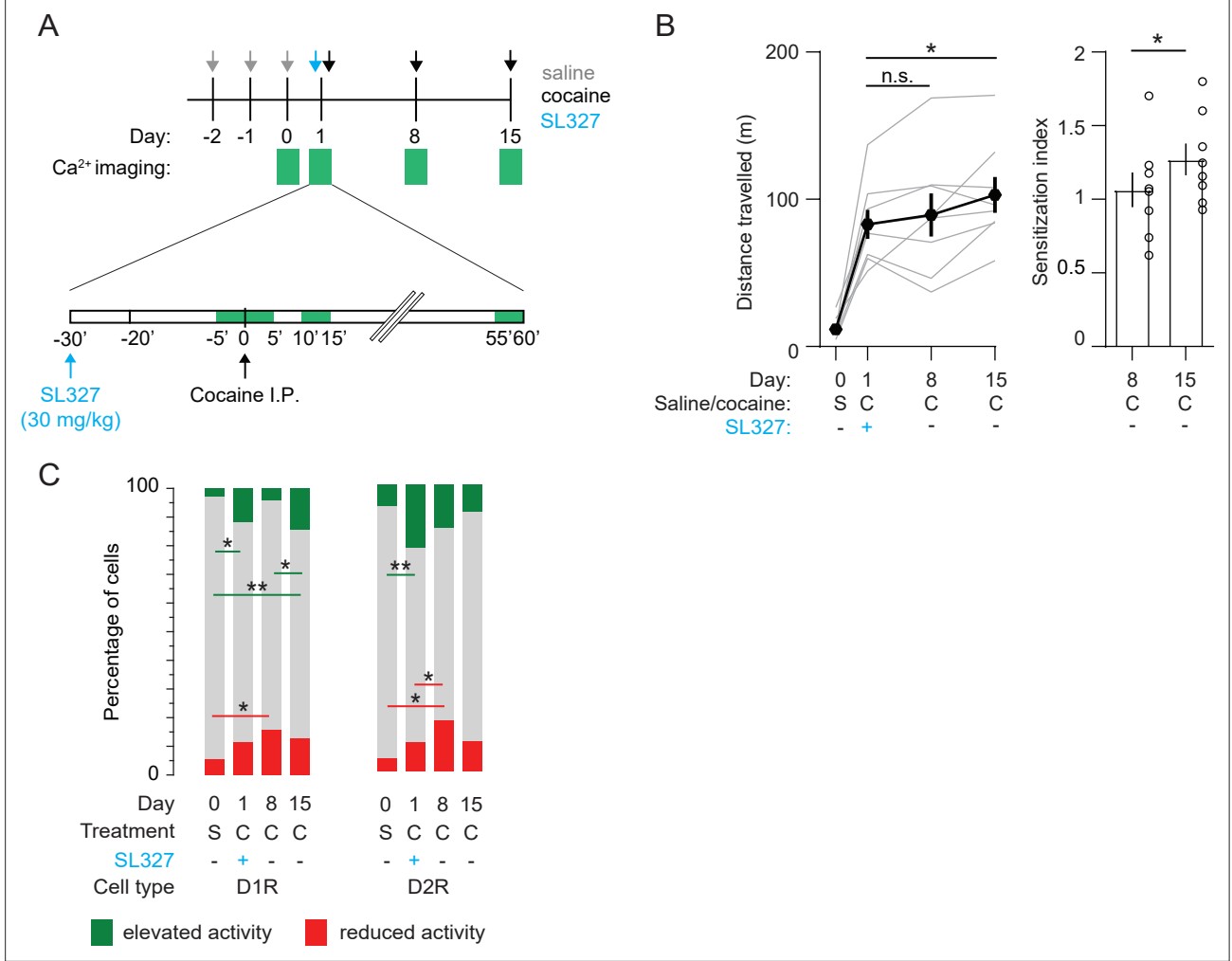

**Figure 6.** Blocking ERK signaling pathway decreases the fraction of sensitized D1R SPN responses, leaves sensitized D2R SPN responses unaffected, and abolishes LS. (**A**) Schematic overview of experimental setup. Animals are pre-treated with SL327 (30 mg/kg) 30 min before the first cocaine injection. Green squares indicate recording days (top) and 5-min recording windows (bottom). (**B**) Left panel: average distance travelled among recorded animals over different days. Paired student T-tests: day 8 vs day 1, p = 0.6, day 15 vs day 1, p = 0.02. Right panel: sensitization index on day 8 versus day 15. Spread shows S.E.M. Students T-test, p = 0.04, N = 8 mice. * = p < 0.05, n.s. = non significant. (**C**) Fraction of significantly responsive D1R and D2R neurons over days. D1R n = 205/192/185/181 neurons (day 0 /day 1 /day 8 /day 15) from 4 mice. Fishers Exact test: elevated activity: day 0 vs day 1: p = 1.6 x 10$^{-2}$, day 8 vs day 15: p = 2.7 x 10$^{-2}$, day 0 vs day 15: p = 5.2 x 10$^{-4}$. Reduced activity: day 0 vs day 8: p = 1.4 x 10$^{-2}$. All other comparisons p> 0.05. D2R n = 141/153/160/176 neurons (day 0 /day 1 /day 8 /day 15) from 4 mice. Fishers exact test, elevated activity: day 0 vs day 1: p = 3.4 x 10$^{-3}$, reduced activity: day 0 vs day 8: p = 4.1 x 10$^{-2}$, day 1 vs day 8: p = 4.5 x 10$^{-2}$. All other intra-group comparisons p > 0.05. * = p < 0.05, ** = p < 0.01. S = saline, C = cocaine.

The online version of this article includes the following figure supplement(s) for figure 6:

**Figure supplement 1.** Blocking ERK signaling pathway prevents D1R sensitization on day 8.

2006; *Pascoli et al., 2011*). To investigate the effects of ERK inhibition on NAc population dynamics, we prepared a new cohort of mice (*Figure 1—figure supplement 1D*). We pretreated these animals with the ERK-inhibitor SL327 (30 mg/kg) 30 min before the first cocaine exposure. This abolished LS one week later, while leaving the acute cocaine-evoked locomotion intact (*Figure 6A and B*). However, since ERK was not inhibited during the second cocaine injection (day 8), we reasoned that sensitization should reappear after a third cocaine injection (day 15). Indeed, we observed that after a third cocaine injection sensitization was partially restored (*Figure 6B*). We then asked whether these behavioral observations were mirrored by activity of NAc SPNs. In D1R SPNs, the fraction of cocaine-activated neurons during the first cocaine exposure, in the presence of the ERK-pathway inhibitor, was still increased compared to saline treatment, like what was observed previously (*Figure 2D*, *Figure 6—figure supplement 1*). However, during the second cocaine injection, when the locomotion was not sensitized, the fraction of cocaine-activated neurons did not increase (*Figure 6C*). For comparison, without SL327 this fraction was larger than during the first cocaine exposure (*Figure 6—figure supplement 1*). Moreover, when animals were treated with cocaine for a third time, locomotor sensitization was restored and the fraction of activated D1R SPNs went up again (*Figure 6C*). This means that during initial cocaine treatment the amount of activated D1R SPNs increases, and the fraction of activated D1R SPNs follows the sensitization-state of the animal. In contrast, in D2R SPNs, the fraction of neurons with reduced activity increases over cocaine exposures, even when LS is blocked by treatment with SL327 (*Figure 6C*). This indicates that while during cocaine exposures the amount of inactivated D2R SPNs increases, their numbers do not reliably track the sensitization-state of the animal. Overall, these results indicate that blocking ERK signaling prevents sensitization, without affecting acute cocaine-induced locomotion. While there is a clear divergence of population response in D1R and D2R SPNs in the NAc during cocaine exposures, the D1R SPNs population activity is causally involved in cocaine locomotor sensitization.

## Discussion

Here, we observe that during LS more D1R SPNs increase their activity, while more D2R SPNs become less active. This indicates that the dichotomy of accumbal population dynamically heightens over cocaine exposures. We find that the dopamine elevations in response to repeated cocaine exposures have similar amplitudes and thus cannot explain the enhanced locomotor response. Inhibition of the ERK-pathway during the first cocaine exposure blocked LS and the recruitment of cocaine-responsive D1R-SPNs. As the increase in D2R-SPNs silencing with repeated cocaine exposures remained unaffected, this leads us to conclude that recruitment of active D1R SPNs eventually drives LS.

Our findings are in line with the observation that cocaine drives c-Fos expression in a subset of D1R SPNs (*Bertran-Gonzalez et al., 2008*), and that the ablation of these c-Fos-expressing neurons is sufficient to block sensitization (*Koya et al., 2009*). Moreover, our data confirm the crucial role of ERK signaling, which acutely did not affect cocaine-induced locomotion or NAc activity patterns. However, it prevented sensitization 1 week later, as well as recruitment of additional D1R SPNs with elevated activity. ERK signaling in D1R SPNs is induced by D1R activation and concomitant NMDA receptor activation (*Valjent et al., 2005*). Moreover, the cAMP-activated guanine nucleotide exchange factor NCS-Rapgef2 plays an important role in this signaling cascade (*Jiang et al., 2021*), eventually causing synaptic potentiation of afferents from the mPFC and ventral hippocampus (*Pascoli et al., 2014*). The first cocaine exposure would lead to widespread potentiation of excitatory afferents, such that more D1R SPNs respond to the second cocaine injection. This recruitment of D1R-SPNs is then sufficient to induce the LS.

What might determine the diversity of response profiles among D1R SPNs? Previous electrophysiological in vivo recordings suggest the existence of distinct functional subclasses within the NAc. Some neurons code for reward and their predictors (*Roitman et al., 2005*), consummatory behavior (*Taha and Fields, 2005*), or action selection (*Nicola, 2007*). Additionally, within the D1R-expressing subclass of SPNs, diversity might be determined by additional molecular markers (*Savell et al., 2020*), or projection specificity. In the dorsal striatum, reducing excitability of striatopallidal or striatonigral projecting SPNs differentially affects LS to amphetamine (*Ferguson et al., 2011*). In analogy, it is possible that the activity of D1R SPNs projecting to midbrain, ventral pallidum (*Kupchik et al., 2015*) or lateral hypothalamus (*O'Connor et al., 2015*) may segregate with movement, incentive saliency or

cocaine response. Understanding the output divergence of accumbal population activity will eventually allow for a circuit-wide understanding of drug-adaptive changes.

We also identified a subpopulation of D2R SPNs, which reduces their activity in response to cocaine and whose number increases during LS. This activity reduction may be due to a reduction of excitatory afferents transmission, potentiated inhibitory transmission, or enhanced G$_{i/o}$-signaling. While the LS protocol used here reliably induces a potentiation on D1R SPNs, no such potentiation is observed on D2R SPNs (*Pascoli et al., 2011*). Moreover, the induction of LTP onto D2R-SPNs requires a decrease of dopamine levels (*Iino et al., 2020*), which is the opposite of the cocaine effect. However, it remains possible that upstream activity is altered during LS, for instance through neuronal subpopulations in the amygdala that specifically target D2R-SPNs (*Shen et al., 2019*).

Reciprocal inhibition between D1R and D2R SPNs seems unlikely to be the major driving force of inhibited D2R SPNs, as D1R SPNs in animals that are treated with an ERK-inhibitor were not activated, but D2R SPNs were still inhibited. In addition, SPN interconnectivity is biased in favor of D2R-SPN inhibition of D1R SPNs (*Taverna et al., 2008*). Potentiation of afferents from local interneurons, or depotentiation of long-range excitatory inputs may contribute to reduced D2R-SPN activity. Alternatively, even in the absence of GIRK channels (*Karschin et al., 1996*), G$_{i/o}$ coupled receptors may silence neurons, for example by inhibition of calcium channels. Whether this also blocks action potentials remains to be determined. In any case, G-coupled signaling cascades can undergo plasticity (*Han et al., 2006*; *Marron Fernandez de Velasco et al., 2015*). It has also been shown that in cortical and habenular neurons, which express GIRKs, cocaine or aversive stimuli weaken the coupling to the effectors (*Hearing et al., 2013*; *Lecca et al., 2016*).

Our data also suggests that LS goes beyond increasing movement. Just as in the dorsal striatum (*Klaus et al., 2017*; *Parker et al., 2018*; *Barbera et al., 2016*), some accumbal SPNs showed strong locomotor alignment. However, the fraction of movement-responsive neurons remained limited and was unchanged with repeated cocaine injections. LS is thus not simply expressed by enhanced accumbal motor coding, but rather by the recruitment of additional cocaine responsive SPNs that may code for increased incentive value (*Berridge, 2007*). As a result, this may favor drug-related associative learning (*Di Chiara, 2002*; *Keiflin and Janak, 2015*) via downstream midbrain targets, enhancing information flow through spiraling connectivity towards more dorsal parts of the striatum (*Haber et al., 2000*). As such, heightened D1-D2 dichotomy may favor the transition to compulsive seeking and taking of addictive drugs. Mechanistic insights into the circuits of early drug-adaptive behavior, such as LS, are essential to establish a comprehensive circuits model, which will enable to draw blueprints for rational therapies for addiction.

## Materials and methods
### Animals
All experiments were reviewed and approved by the Institutional Animal Care and Use Committee of the University of Geneva (No GE128-16, GE64-20 and GE71-20). Adult male and female wildtype C57/Bl6 (Charles River) mice, and transgenic heterozygote mice for either D1R-Cre (Cre expressed under the control of the *Drd1a* promoter, B6.FVB(Cg)-Tg(Drd1a-cre)EY262Gsat/Mmucd, identification number 030989-UCD), D2R-Cre (Cre expressed under the control of the *Drd2* promoter, B6.FVB(Cg)-Tg(Drd2-cre)ER44Gsat/Mmucd, identification number 032108-UCD) or A2A-Cre (Cre expressed under the control of the *Adora2a* promoter B6.FVB(Cg)-Tg(Adora2a-cre)KG139Gsat/Mmucd, identification number 036158-UCD) were used for all experiments. Animals were kept in temperature- and hygrometry-controlled environments with a 12 hr light / 12 hr dark cycle. All experiments were performed in the light phase.

### Viral injections and chronic implants
For miniaturized microscopy imaging, D1R-cre, D2R-cre and A2a-cre transgenic mice were injected with an AAV encoding the calcium indicator GCaMP6f (AAV5-hSyn-FLEX-GCaMP6f, University of Pennsylvania, Vector Core) into the Nucleus Accumbens ( + 1.7 mm AP, ± 0.7 mm ML, –4.4 mm DV from bregma), or dorsal striatum ( + 1.4 mm, ± 1.5 mm, –2.4 mm, respectively). For fiber photometry recordings, wildtype mice were injected into the NAc with an AAV encoding the dopamine sensor dLight (AAV5-CAG-dLight1.1, Addgene). Afterwards a gradient refractive index (GRIN) lens (0.5 mm

or 0.6 mm diameter for NAc, 1.0 mm diameter for dStr, Inscopix) or an optic fiber (0.4 mm diameter, MFC_400/430–0.48_4 mm_ZF2.5(G)_FLT, Doric Lenses) was placed above the viral injection site and secured to the skull using dental cement. After 4–6 weeks, a baseplate that holds the miniscope (nVista 2.0 or nVista 3.0, Inscopix) was attached to the implant.

## Locomotor sensitization

Locomotor activity was recorded in small circular open fields (18 cm diameter). Animals were initially habituated for a few days to the recording room as well as to handling, fixation, and miniaturized microscope connection on the baseplate. For the next 2–3 days they were also habituated to the open field, as well as to intraperitoneal injections (by injecting saline 10 mL/kg). On subsequent recording days, animals were attached to the microscope or optic fiber, habituated for 10 min in their home cages, and were then placed in the open field for 20 min before receiving an intraperitoneal injection of either saline or cocaine (20 mg/kg, both injections at 10 mL/kg). In case of SL327 manipulations, animals were injected with SL327 on the day of the first cocaine-exposure day, immediately after attaching the microscope (30 min before cocaine injection). Animals' movements in the locomotor arena were recorded by a camera (C922 Pro, Logitech) at 25–30 Hz. Animals' saline and cocaine loco-motion responses were quantified using Anymaze software (Stoelting).

## Calcium signal recordings

Calcium recordings were performed using an nVista miniaturized epifluorescent microscope (Inscopix). All recordings were made at 10 frames per second. Optimal field of view was determined during habituation 1–2 days before starting the recordings. LED intensities (0.1–2 mW) and gain (1–5 x) were optimized per animal for sufficient signal-to-noise while minimizing fluorescent bleaching. Timing of each acquired frame (SYNC-port on Inscopix data acquisition box) was synchronized with behavioral recordings through sending TTLs to an Arduino Uno (Arduino).

## Fiber photometry recordings

The fiber photometry recordings were performed like earlier reports (*Corre et al., 2018*). Excitation (M470F3, Thorlabs) and control (405 nm, M405FP1, Thorlabs) LED light was sinusoidally modulated and passed through excitation filters onto an optic fiber patch cable that interfaced with the animals' chronically implanted fiber (MFP_400/430/1100–0.48_4 m_FC-ZF2.5, Doric Lenses). Emission light travelled through the same fiber onto a photoreceiver (Newport 2151, Doric Lenses), after which it was demodulated and stored using a signal processor (RZ5P, Tucker Davis Technologies). Before recording drug induced DA transients, mice were habituated with saline I.P. injections and handling for 5 days. On the following days, mice were connected to the fiber photometry set up and recorded for 5 min as baseline, followed by I.P. injections, and at least 15 min recordings afterwards.

## Drugs

All drugs were injected at 10 mL/kg. Cocaine-HCl (obtained from the University Hospital of Geneva) was dissolved in saline and injected at 20 mg/kg. SL327 (Sigma-Aldrich) was dissolved in DMSO 12.5 % (vol/vol) in water and injected at 30 mg/kg. Raclopride (Tocris) was dissolved to 0.3 mg/mL in saline containing 10 % (2-hydroxypropyl)-β-cyclodextrin (Tocris), sonicated, and diluted to 0.01 mg/mL for injection.

## Histology

Following behavioral experiments, animals were deeply anesthetized using pentobarbital (150 mg/kg), after which they were transcardially perfused with 4 % formaldehyde in PBS. Brains were stored for 24 hr in 4 % formaldehyde and then transferred to PBS. They were then cut to 50–60 μm slices on a vibratome (Leica VT1200S), and DAPI stained. Placement of lens/fiber, as well as virus expression, were confirmed (*Figure 1—figure supplement 1*).

## Imaging processing and neuron detection

The acquired images were spatially downsampled 4 x, cropped, and motion corrected using Mosaic or Inscopix Data Processing Software software (Inscopix). Identification of regions of interest (ROIs) was then done using Constrained Nonnegative Matrix Factorization for microEndoscopic data (CNMF-E),

a method to 'accurately separate the background and then demix and denoise the neuronal signals of interest' (*Zhou et al., 2018*). We used the CNMF-E tools neuron.merge, neurons_dist_corr(), neuron.merge_hig_corr(), neuron.merge_close_neighbors and neuron.quickMerge, to automatically remove potential noise ROIs. To further improve the quality of the isolation, manual quality control was performed by inspecting the spatial and temporal data for each ROI, using CNMFE 'A' and CNMFE 'C_Raw' to extract the spatial and temporal profiles of the ROIs. We manually checked all pairs of ROIs for which the raw temporal traces were correlated ($R > 0.6$). In addition, ROIs with non-standard calcium dynamics (i.e. no fast rise and slower decay) were excluded, as they were regarded as possible (a) overexpressing neurons (b) fluorescent debris moving in and out of focus or (c) possible interneurons that express either D1R or D2R (*Gritton et al., 2019*). ROIs with poor signal/noise quality were also excluded. Furthermore, we identified the total surface area of each ROI and checked the smallest ROIs manually to assess whether or not they could be considered as neurons. Following quality control, the temporal ('C_raw' label in CNMF-E) and spatial ('A' label in CNMF-E) data were used for further analyses.

## Fiber photometry analysis

Demodulated 405 nm originating and 470 nm originating signals were used for analysis. To calculate dF/F, the 405 signal was linearly regressed to the 470 signal, and then subtracted to create dF/F (470 nm signal – fitted 405 nm signal)/ fitted 405 nm signal. We then averaged dF/F to the appropriate time bins used in the graphs and analysis.

## Transient detection

In order to identify individual transients a custom code was developed that identifies calcium transients by their temporal dynamics (fast rise, slow decay), and then fits the individual transients to estimate the amplitude, area, and onset time of individual transients. Briefly, from the CNMF-E 'C_raw' output, a moving average of the lowest 20 % of values was subtracted in a sliding window to correct for slow variations of the signal. The derivative of the signal was computed and thresholded to detect prolonged periods of increasing and decreasing signal, indicative of onset and offset of potential transients. Next, we isolated increase-decrease epoch pairs to fit the individual transients (1): the fast increase is modeled by a generalized logistic function with three parameters ($\alpha_p$ : upper asymptote, $\tau_g$ : growth rate, $t_g$ : time of maximum growth) while the slow decay is modeled by an exponential function with two parameters ($\tau_d$ : decay rate, $t_d$ : time of the decay). We take the product of these two functions and ensure their match by imposing $t_g = t_d = t_0$ . Thus, we get the following functional form with four parameters:

$$f_{peak}(t) = \frac{\alpha_p}{1+e^{(t-t_0)/\tau_g}} * e^{-(t-t_0)/\tau_d},$$
(1)

This function is then used to perform a non-linear least-square fitting. Initially, the average transient trace was fitted to determine reference values of the parameters ($\alpha_p$ , $\tau_g$ , $\tau_d$, and $t_0$) before fitting the individual transients. After having fitted each isolated transient, the data was cleaned by removing fits with abnormal shape, or low amplitudes (relative to time periods with no transients). Finally, from these fitted individual transients the continuous trace was reconstructed and estimates were made of the relative contributions of individual transients. For later analysis, the transient half-rise time was used as a proxy for calcium transient onset.

## Identifying cocaine-responsive neurons

To identify cocaine-responsive neurons a bootstrapping resampling method was used (*Rozeske et al., 2018*). Per neuron, the change in calcium transient frequency was calculated between the last 5 min before the injection to the first 5 min after the injection. We then took all inter transient intervals of that particular neuron, and created 10,000 shuffled distributions of the intervals. From this shuffled data the transient frequency activity change between the same time periods was again calculated. Neurons with activity changes that were bigger than the 99th percentile value of the shuffled data were considered to be neurons with increased activity, while neurons with activity changes lower than the 1st percentile value of shuffled data were considered to be neurons with decreased activity.

## Locomotor analysis during calcium imaging

To align the calcium imaging signal with animals' movements, we identified each animal's frame-by-frame movement using MATLAB (Mathworks), similarly to what has been done before (*Ziv et al., 2013*). Individual video frames were binarized, and the Euclidean distance between the center points of the animal was calculated. Data was smoothed (0.5 s sliding average), then aligned and averaged per calcium imaging frame (10 Hz). To identify movement-aligned neurons, a similar approach as the method described above for identifying cocaine-responsive neurons was used. For each neuron, the movement (–5 s to +5 s) was aligned around each calcium transient. To identify movement-changes around calcium transients, the baseline activity (average of –5 to –3 s) was subtracted. Next, the movement speed around calcium transient firing (–0.5 to +0.5 s around calcium transient) was averaged over all calcium transients from a single neuron. Per neuron we compared the movement speed in this window around the calcium transient to 10,000 shuffled data where the movement speed was shifted by 10–1200 s. Movement-related neurons were defined as neurons for which the non-shifted data was among the top five percent of the shuffled controls.

## Statistical tests and experimental considerations

Statistical analyses were either performed in Matlab or Prism (Graphpad). All statistical tests were corrected for multiple comparisons with the Bonferroni method whenever applicable. Behavioral movement data was assessed using parametric statistical tests (repeated measures ANOVA followed by post hoc students paired/unpaired t-tests). Identification of cocaine or movement-responsive neurons was done as described above. Comparisons of fractions of responding neurons were done using Fisher's exact test, and other comparisons of single neuron activity was done using the non-parametric Wilcoxon signed-rank test. As there was no pre-defined effect size, the group sizes were chosen to match those of miniaturized epifluorescence single neuron calcium imaging studies in the dorsal striatum (*Klaus et al., 2017*; *Parker et al., 2018*; *Barbera et al., 2016*). Animals were assigned to test/control groups in a random fashion, and experiments were run in small cohorts of two to four animals per recording day. Data acquisition and analysis was done in a non-blinded fashion. Animals in which < 20 neurons were recorded were excluded from the analysis.

## Linear mixed model

The response of each neuron to injection was quantified as the following ratio: log2((number of transients after injection +10)/(number of transients before injections + 10)). The log2 ratio was applied to approach normality, and the constant 10 was added to mitigate the effect of small values. We used the R package lme4 (v1-27.1) to construct linear mixed models of the log2 ratio as a function of time or cell type, and individual animals as random effect. Tukey post-hoc tests were performed with the package emmeans (v1.6.2–1).

## Acknowledgements

This work is dedicated to Jérôme Flakowski who passed on 11.5.2021, and whose contributions to this project were crucial. We thank the members of the Lüscher lab for the valuable discussions and thank Vincent Pascoli and Anthony Holtmaat for their comments on earlier versions of the manuscript. CL is supported by the Swiss National Science Foundation and the European Research Council (ERC).

## Additional information

### Funding

| Funder | Grant reference number | Author |
| --- | --- | --- |
| Swiss National Science Foundation | 310030_189188 | Christian Lüscher |
| European Commission | F_Addict | Christian Lüscher |

| Funder | Grant reference number | Author |
|---|---|---|
| Swiss National Science Foundation | CRSII5_186266 | Christian Lüscher |

The funders had no role in study design, data collection and interpretation, or the decision to submit the work for publication.

## Author contributions

Ruud van Zessen, Conceptualization, Data curation, Formal analysis, Investigation, Methodology, Resources, Software, Visualization, Writing - original draft, Writing - review and editing; Yue Li, Investigation, Validation; Lucile Marion-Poll, Data curation, Formal analysis, Writing - review and editing; Nicolas Hulo, Formal analysis, Methodology, Software; Jérôme Flakowski, Data curation, Formal analysis, Software, Visualization, Writing - review and editing; Christian Lüscher, Conceptualization, Funding acquisition, Project administration, Supervision, Writing - original draft, Writing - review and editing

## Author ORCIDs

Ruud van Zessen (iD) http://orcid.org/0000-0001-5634-6922
Lucile Marion-Poll (iD) http://orcid.org/0000-0003-4652-9797
Nicolas Hulo (iD) http://orcid.org/0000-0003-2640-636X
Jérôme Flakowski (iD) http://orcid.org/0000-0002-6457-3022
Christian Lüscher (iD) http://orcid.org/0000-0001-7917-4596

## Ethics

All experiments were reviewed and approved by the Institutional Animal Care and Use Committee of the University of Geneva (GE-128-16, GE64-20 and GE71-20).

## Decision letter and Author response

Decision letter https://doi.org/10.7554/eLife.66048.sa1
Author response https://doi.org/10.7554/eLife.66048.sa2

## Additional files

### Supplementary files
• Transparent reporting form

### Data availability

Data and code have been made available via Zenodo: DOI:https://doi.org/10.5281/zenodo.5507009.

The following dataset was generated:

| Author(s) | Year | Dataset title | Dataset URL | Database and Identifier |
|---|---|---|---|---|
| van Zessen R, Li Y, Marion-Poll L, Hulo N, Flakowski J, Lüscher C | 2021 | Dynamic dichotomy of accumbal population activity underlies cocaine sensitization | https://doi.org/10.5281/zenodo.5507009 | Zenodo, 10.5281/zenodo.5507009 |

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
