## [Decision Letter]

**Acceptance summary:**

This study uses single cell calcium imaging to examine the activity dynamics of nucleus accumbens (NAc) spiny projection neurons expressing dopamine receptor 1 (D1 neurons) or dopamine receptor 2 (D2 neurons) in response to acute cocaine and after sensitization. It finds that both populations contain neurons that are inhibited or excited by acute cocaine, but that sensitization may alter the proportions of responsive neurons in distinct ways, increasing proportions of excited D1 neurons and inhibited D2 neurons after sensitization. These results shed light on the cellular basis of cocaine-induced locomotor sensitization.

**Decision letter after peer review:**

Thank you for submitting your article "Dynamic dichotomy of accumbal population activity underlies cocaine sensitization" for consideration by *eLife*. Your article has been reviewed by 2 peer reviewers, and the evaluation has been overseen by a Reviewing Editor and Michael Taffe as the Senior Editor. The reviewers have opted to remain anonymous.

Essential revisions:

1) The support for the main conclusions would be strengthened by the revisions or additions to the statistical analyses:

– Direct comparisons between the relevant conditions (D1 versus D2 neurons, SL327 versus untreated controls), rather than separate tests in each condition that are reported as either significant or non-significant (since the absence of a significant effect isn't evidence for the absence of any effect). In some cases, the data are changing in opposite directions, making it very likely that these direct comparisons will be significant, but in other cases it is less clear.

– Analyses that account for the lack of independence between neurons recorded from the same subjects. This can be tricky when examining binary/proportion data (I'm not aware of a standard equivalent to a Fisher's exact test that accounts for lack of independence) – I would suggest calculating the proportion of neurons that are excited or inhibited for each subject individually and then running a mixed or repeated measures ANOVAs to assess whether these proportions are changing over time. For continuous measurements like change in transients, tests like linear mixed models with random effects for subjects could be used to determine whether differences in D1 or D2 activity changes were consistent across subjects.

2) Discussion of the results in the context of the following papers would strengthen the manuscript:

– Ferguson et al., Nat. Neurosci. 2011 Transient neuronal inhibition reveals opposing roles of indirect and direct pathways in sensitization.

– Jiang et al., JNeurosci 2021 Cocaine-Dependent Acquisition of Locomotor Sensitization and Conditioned Place Preference Requires D1 Dopaminergic Signaling through a Cyclic AMP, NCS-Rapgef2, ERK, and Egr-1/Zif268 Pathway – this paper just came out but it would be informative for the authors to include in their discussion.

3) The data in Figure 5 are used to conclude that NAc dopamine was stable during cocaine sensitization. Could this be due to a ceiling effect? That is, if the first cocaine injection elicited a maximal sensor response, it would be incorrect to conclude that a second dose does not increase dopamine release. While additional experiments are not requested at this time, this was a significant concern discussed in the review process and a thoughtful discussion of this potential caveat should be included in the Discussion section.

4) To what degree can epifluorescence measurements be used to assign calcium activity to individual neurons? The authors cite Zhou et al., (bioRxiv, but it is now published in *eLife* so this reference should be ipdated) and use constrained nonnegative matrix factorization to assign ROIs for putative somata. There was concern by the reviewers that this isn't enough to evaluate the "isolation quality" (e.g., the morphologies in Figure 1D and Figure 3A look strange). The authors are asked to make it clear to the reader how to access their raw data through GitHUB or a related mechanism which could help to understand the path from raw to processed data.

*Reviewer #1 (Recommendations for the authors):*

In this manuscript, van Zessen and colleagues use single cell calcium imaging to examine the activity dynamics of nucleus accumbens (NAc) spiny projection neurons expressing dopamine receptor 1 (D1 neurons) or dopamine receptor 2 (D2 neurons) in response to acute cocaine and after sensitization. They report that both populations contain neurons that are inhibited or excited by acute cocaine, but that sensitization may alter the proportions of responsive neurons in distinct ways. Specifically, they reported increasing proportions of excited D1 neurons and inhibited D2 neurons after sensitization. They also report that dopamine responses to cocaine are stable and do not increase with sensitization and that blocking ERK signaling prevent sensitization (as reported previously) and prevent the increase in excited D1 neurons. The experiments are technically rigorous and include appropriate controls. Many of the analyses are rigorous, but some of the statistical approaches used to support the main conclusions are weak or inappropriate.

Strengths:

– The in vivo calcium activity patterns of individual D1 and D2 NAc neurons in response to acute cocaine and after locomotor sensitization have, to my knowledge, not been reported before, making this a valuable data set.

– Rigorous methods were used for identifying cocaine-responsive neurons, detecting transients, identifying movement-related neurons, and assessing the role of movement-encoding in sensitization-related activity pattern changes.

Weaknesses:

– The responses of D1 and D2 neuron populations are not directly statistically compared; instead, D1 responses to cocaine are compared to saline, and similarly for D2 responses. This weakens some of the main conclusions are about differences between these populations.

– Most of the analyses used to support the main conclusions (Fisher's exact test, non-parametric Wilcoxon signed-rank test) rely on the assumption that the measurements are independent, whereas individual neurons measured in the same animal are not independent. Therefore, some of the differences reported here could be due to random effects of subject that are not accounted for.

– The evidence for the role of activity changes in a subgroup of D1 neurons in locomotor sensitization, as described in abstract, is indirect. The ERK-inhibition SL-327 appears to be prevent locomotor sensitization, and to prevent the recruitment of additional cocaine-excited neurons, but the SL-327 effects on locomotor sensitization could be related to other effects on population activity, or mechanisms that are not reflected in cocaine-induced activity changes.

– The effects of SL-327 on neural and behavioral changes from cocaine day 1 to day 8 are not directly compared to an untreated control group. Instead, they are compared to subsequent untreated sensitization on day 15 (which is not assessed in animals that did not receive SL-327).

The support for the main conclusions would be strengthened by the revisions or additions to the statistical analyses:

– Direct comparisons between the relevant conditions (D1 versus D2 neurons, SL327 versus untreated controls), rather than separate tests in each condition that are reported as either significant or non-significant (since the absence of a significant effect isn't evidence for the absence of any effect). In some cases, the data are changing in opposite directions, making it very likely that these direct comparisons will be significant, but in other cases it is less clear.

– Analyses that account for the lack of independence between neurons recorded from the same subjects. This can be tricky when examining binary/proportion data (I'm not aware of a standard equivalent to a Fisher's exact test that accounts for lack of independence) – I would suggest calculating the proportion of neurons that are excited or inhibited for each subject individually and then running a mixed or repeated measures ANOVAs to assess whether these proportions are changing over time. For continuous measurements like change in transients, tests like linear mixed models with random effects for subjects could be used to determine whether differences in D1 or D2 activity changes were consistent across subjects.

Discussion of the results in the context of the following papers would strengthen the manuscript:

– Ferguson et al., Nat. Neurosci. 2011 Transient neuronal inhibition reveals opposing roles of indirect and direct pathways in sensitization.

– Jiang et al., JNeurosci 2021 Cocaine-Dependent Acquisition of Locomotor Sensitization and Conditioned Place Preference Requires D1 Dopaminergic Signaling through a Cyclic AMP, NCS-Rapgef2, ERK, and Egr-1/Zif268 Pathway – this paper just came out but it would be informative for the authors to include in their discussion.

*Reviewer #2 (Recommendations for the authors):*

van Zessen et al., measured calcium dynamics of two cell types in dorsal and ventral striatum in mice exposed to two doses of cocaine. They show opposite responses of D1R and D2R SPNs following cocaine. Changes in D1R neurons depended on intact ERK signaling, while dopamine transients appeared not to account for the measured changes. Overall, the results are relatively straightforward and will contribute to knowledge of the acute effects of cocaine on locomotor sensitization. My main concern is whether the data strongly support the conclusions. In particular, I have questions about the ability to measure fluorescence from individual neurons and potential ceiling effects of dopamine measurements.

1. The data in Figure 5 are used to conclude that NAc dopamine was stable during cocaine sensitization. Could this be due to a ceiling effect? That is, if the first cocaine injection elicited a maximal sensor response, it would be incorrect to conclude that a second dose does not increase dopamine release.

2. To what degree can epifluorescence measurements be used to assign calcium activity to individual neurons? The authors cite Zhou et al., (bioRxiv, but I believe it's been published in *eLife*) and use constrained nonnegative matrix factorization to assign ROIs for putative somata. This isn't enough to evaluate the "isolation quality" (e.g., the morphologies in Figure 1D and Figure 3A look strange). It would also be important to see more raw data to help the reader understand the path from raw to processed data.

3. Were there spatial correlations in the direction and magnitude of responses to cocaine in D1R and D2R neurons? Previous work in mouse striatum has shown spatial organization (Barbera et al., Neuron, 2016; Klaus et al., Neuron, 2017) in other behavioral domains.

---

## [Author Response]

Essential revisions:1) The support for the main conclusions would be strengthened by the revisions or additions to the statistical analyses:– Direct comparisons between the relevant conditions (D1 versus D2 neurons, SL327 versus untreated controls), rather than separate tests in each condition that are reported as either significant or non-significant (since the absence of a significant effect isn't evidence for the absence of any effect). In some cases, the data are changing in opposite directions, making it very likely that these direct comparisons will be significant, but in other cases it is less clear.

We thank the reviewers for this constructive suggestion. Direct comparisons between the two populations are indeed a powerful way to make the case of cell type dichotomy. We now directly compare the response to cocaine in D1R and D2R SPNs (Figure 2D), also when SL327 was applied (Figure 6_S1). We find significant differences, thus strengthening our conclusions.

– Analyses that account for the lack of independence between neurons recorded from the same subjects. This can be tricky when examining binary/proportion data (I'm not aware of a standard equivalent to a Fisher's exact test that accounts for lack of independence) – I would suggest calculating the proportion of neurons that are excited or inhibited for each subject individually and then running a mixed or repeated measures ANOVAs to assess whether these proportions are changing over time. For continuous measurements like change in transients, tests like linear mixed models with random effects for subjects could be used to determine whether differences in D1 or D2 activity changes were consistent across subjects.

We agree with the reviewer that non independence of data from the same animal needs to be accounted for in the analysis. We compared the number of transients before and after cocaine injections by calculating the log2(ratio after/before) and found a normal distribution. We then ran a linear mixed model with animals as random effects, as suggested by the reviewers. Finally, we have also quantified the variance per animal by plotting sdev of the calcium transients. These analyses are now presented in Figure S2_S1 and described in the Methods; the results agree with our previous analysis on the proportions.

2) Discussion of the results in the context of the following papers would strengthen the manuscript:– Ferguson et al., Nat. Neurosci. 2011 Transient neuronal inhibition reveals opposing roles of indirect and direct pathways in sensitization.– Jiang et al., JNeurosci 2021 Cocaine-Dependent Acquisition of Locomotor Sensitization and Conditioned Place Preference Requires D1 Dopaminergic Signaling through a Cyclic AMP, NCS-Rapgef2, ERK, and Egr-1/Zif268 Pathway – this paper just came out but it would be informative for the authors to include in their discussion.

We agree with the reviewers that these papers provide valuable insights into molecular machinery in circuits of the dorsal striatum underlying locomotor sensitization. We have now added both references to the discussion.

3) The data in Figure 5 are used to conclude that NAc dopamine was stable during cocaine sensitization. Could this be due to a ceiling effect? That is, if the first cocaine injection elicited a maximal sensor response, it would be incorrect to conclude that a second dose does not increase dopamine release. While additional experiments are not requested at this time, this was a significant concern discussed in the review process and a thoughtful discussion of this potential caveat should be included in the Discussion section.

We agree with the reviewers to ensure that the fluorescent signal is not saturated. We therefore performed an additional experiment in a new batch of wildtype animals that we injected with an AAV containing dLight into the NAc. We then monitored dopamine in response to with 20 mg/kg cocaine IP as before, but then followed up with an injection of the D2 antagonist raclopride 0.1 mg/kg IP. The injection was timed to the peak of the cocaine response (see figure 5B). We reasoned that this should relieve the autoinhibition of DA neurons by D2R receptor activity (Wei et al., Cell Discovery 2018), thus further increasing DA levels. We indeed observed a significant enhancement of the dLight fluorescence, confirming that cocaine did not saturate the indicator. These results are presented as a new panel in Figure 5 and are described in the Results section.

4) To what degree can epifluorescence measurements be used to assign calcium activity to individual neurons? The authors cite Zhou et al., (bioRxiv, but it is now published in eLife so this reference should be ipdated) and use constrained nonnegative matrix factorization to assign ROIs for putative somata. There was concern by the reviewers that this isn't enough to evaluate the "isolation quality" (e.g., the morphologies in Figure 1D and Figure 3A look strange). The authors are asked to make it clear to the reader how to access their raw data through GitHUB or a related mechanism which could help to understand the path from raw to processed data.

Indeed, single photon epifluorescence calcium imaging carries the risk of picking up noise from neighboring neurons, and we did carefully consider this in the data processing pipeline. We have now updated the Methods section “Image processing and neuron detection”, giving in depth details on the path from raw to processed data. All the raw data and procedures will be uploaded to Zenodo (CERN, Geneva), for which the link is in the manuscript.

The examples of the fields of view in Fig1D and Figure 3A might “look strange”, but this is only due to an edge distortion typical for GRIN lenses.

In addition, the citation mentioned has been updated.